# Numerical Optimization Simulation of Synchronous Four-Wing Rotor

**DOI:** 10.3390/ma13235353

**Published:** 2020-11-25

**Authors:** Kongshuo Wang, Haichao Liu, Tianhao Chang, Deshang Han, Yiren Pan, Chuansheng Wang, Huiguang Bian

**Affiliations:** 1College of Electromechanical Engineering, Qingdao University of Science and Technology, Qingdao 266061, China; kongshuo726@163.com (K.W.); liuhaichao66mm@qust.edu.cn (H.L.); chang_tianhao@163.com (T.C.); 17853253362@163.com (D.H.); pyr@qust.edu.cn (Y.P.); 2Academic Division of Engineering, Qingdao University of Science & Technology, Qingdao 266061, China; 3Shandong Provincial Key Laboratory of Polymer Material Advanced Manufactorings Technology, Qingdao University of Science and Technology, Qingdao 266061, China

**Keywords:** mixing, rotor, configuration, PSO optimization, numerical simulation

## Abstract

The mixer is the most widely used batch mixing equipment in the rubber industry. The rotor is a core component and has a great impact on the mixing effect of the equipment. The current rotor structure design is done empirically, being tightly dependent on practical experience. This paper proposes a method for optimizing the rotor structure by using optimization algorithms combined with numerical simulation technology. Using MATLAB software, a parametric design program for synchronous rotors and a set of optimization programs for the particle swarm optimization (PSO) algorithm were written. The global distribution index was used as the fitness function to optimize the synchronous rotor configuration. A comparative analysis of the rotors before and after optimization shows that the optimization process is feasible, and the results are reliable. This provides new ideas for the design and development of mixer rotors.

## 1. Introduction

Rubber is a key material for the auto, medical, and mechanical industries, and it is mainly used to make tires, seals, and hoses, among other components. The two most commonly used methods for obtaining rubber include direct extraction from rubber trees [1] and artificial synthesis. Before the rubber is processed into final products, the raw material goes through key processes that affect the quality of the final product. The first step in rubber processing is mixing, which consists of the addition of active fillers or other polymers to improve its physical properties [2]. The ultimate purpose of mixing is to achieve the refinement and uniform distribution of polymers and additives in a multi-component system through internal mixing equipment so that the various additives are completely and uniformly distributed in the rubber [3]. Such a process takes place in the mixer [4]. The rotor is the core equipment of the mixer, and its good performance is key for obtaining high-quality compounding rubber [5]. Many factors affect the mixing performance, such as the shape of the mixing chamber, the shape and speed of the rotor, and the filling factor, among others [6,7]. In order to achieve the best mixing effect on the filler, it is important to study the rotor configuration and the flow mode of the mixing material in the mixing chamber [7]. The mixing of the rubber compound can be simulated, thanks to the continuous development of the finite element method, numerical simulation technology, three-dimensional drawing technology, and experimental detection technology [8]. Such methods provide new technical means for developing new rotors, optimizing existing rotors, and exploring the feasibility of new mixing processes [9]. Based on this, this paper establishes a rubber mixing model by a finite element simulation method and optimizes the rotor structure with the particle swarm optimization algorithm, obtaining a rotor structure with a better distributed mixing effect. A series of comparative analyses were performed between the optimized and unoptimized rotor. This study is of great significance to further improving the mixing quality and efficiency, shortening the design time of the mixer rotor structure, and developing new mixing components.

For example, Kim J.K. et al. [10] developed a numerical model for isothermal Newtonian fluids in a mixer through slip theory and solved the corresponding equations of the model using the finite difference method. Gramann et al. [11] used the boundary element method to simulate the non-isothermal flow behavior of polymer materials in mixers and accurately predicted the mixing quality of the blend and the viscous heat of the polymer during the mixing process. Hu et al. [12] used a theoretical model for hydrodynamic lubrication to conduct a comparative study of the pressure, velocity, and viscosity field of a four-wing and two-wing rotor of an internal mixer through numerical simulation calculations. Although the method is more accurate and effective, it is only applicable to the analysis of the flow behavior of Newtonian fluids in the mixer under full filling conditions. Alsteens et al. [13] used POLYFLOW (version 19.0) software to simulate and compare the effect of dispersion mixing and distribution mixing in the presence of wall slip and wall slip-free boundary conditions and carried out related experimental studies. The results showed that when considering wall slip, the numerical simulation and experimental results were in good agreement. Connelly et al. [14] used the finite element method, grid overlap technology, and particle trajectory tracking technology to simulate and analyze the rubber mixing process, and obtained a relationship between the viscosity of the rubber material, the shear rate, and other mixing performance characterization parameters as a function of time. Laser Doppler velocimetry experiments were performed to verify the rheological characteristics of shear thinning of rubber in an internal mixer. Moreover, Mohammad Ebrahimpour et al. [15] used the finite volume method to investigate the effect of overlap ratios and the position of blades on a vertical axis wind turbine. Two-dimensional numerical simulations were performed using URANS equations and the sliding mesh method. Rezvan Alamian et al. [16] used a multi-objective genetic algorithm to optimize the shape of the wave energy converter, and the NEMOH (version 2.03) software was used to simulate the wave-body interaction. The results showed that the bottom flat and upside chamfered geometry with X:Y ratio of 10:1 was the best geometry for the desired application.

The particularity of the structure of the synchronous rotor mixer, the viscoelastic behavior of the different compounds, and the contingency in the rubber mixing process make the establishment and calculation of mathematical models very difficult. At present, research on shear rotors is very detailed, but most of them stay at the stage of theoretical analysis, especially in the design process of the rotor rib structure. The novelty of this research lies in the combination of particle swarm optimization (PSO) algorithm and numerical simulation technique to optimize the rotor structure parameters, and the corresponding experiments were carried out to verify the method.

## 2. Numerical Simulation and Optimization Model of Mixing Flow Field

### 2.1. Physical Model

Figure 1 shows the design steps of the parametric model of a synchronous rotor. A series of design parameters are set to program control, which avoids the repeated modeling of the rotor during the optimization process, saving modeling time. The meaning of the parameters in the parameter control editor is as follows: DD—Maximum swivel circle diameter; L—Total axial length; L1—Short-wing length; L2—Long-wing length; D—Base circle diameter; S—Rotor wing width; α—helical angle of short-wing; β—helical angle of long-wing; R1—front surface radius; R2—back surface radius.

By ticking the parameters in the parameter control editor, the values were entered, and the rapid establishment of the 3D model was completed. Because this paper mainly discusses the effect of the long and short wing structure of the synchronous rotor on mixing, some parameters were assigned to ensure the reliability of the comparison. The specific size design parameters are shown in Table 1.

### 2.2. Basic Assumptions

The main task of this simulation is to optimize the global distribution index of the mixing process. MATLAB (version 2018a) and POLYFLOW (version 19.0) software were used. More specifically, the flow field finite element simulation and mixed task numerical simulation were performed in POLYFLOW software, which is time-consuming. Therefore, considering computer simulation speed and accuracy, the following assumptions were made for the finite element and numerical simulation models [17]:The compound is fully filled in the mixing chamber;The compound is isothermal, and the temperature of each point in the flow field is consistent;Rubber flow is laminar, with a small Reynolds number;The gravity and inertia of the rubber are much smaller than the viscosity, and can be ignored;No slippage on the fluid wall;The compound is a power-law fluid, which meets the characteristics of a non-Newtonian fluid.

### 2.3. Mathematical Equation

#### 2.3.1. Time-Varying Governing Equation

For time-varying fluid flow, the governing equation of POLYFLOW can be expressed as follows [18]:(1)$$M\left(X\right)\overset{•}{X}+K\left(X\right)X+F\left(X\right)=0.\;$$

Using the initial conditions:(2)$$X\left(t_{0}\right)=X_{0}a.\;$$

The specific meaning of each unknown in Equations (1) and (2) is as follows:

X-Unknown vector, such as speed, pressure, temperature, viscoelastic additional stress; $\overset{•}{X}$- time-derivative of X;

M-Quality matrix, depends on unknown vector X;

K-Stiffness matrix, depends on unknown vector X;

F-Corresponding volume forcing function and natural boundary conditions.

ANSYS POLYFLOW generally uses a parabolic time-stepping method to solve Equation (1). POLYFLOW replaces the solution set of continuous time by calculating the solution of Equation (1) in discrete time. Its specific expression is as follows [19]:(3)$$X_{n}=X\left(t_{n}\right),\;$$
(4)$$t_{n}=t_{n-1}+{\Delta t}_{n}.\;$$

The subscript n represents the time step.

$\overset{•}{X}$ can be solved inversely by Equation (1):(5)$$f\left(X\right)=-M^{-1}\left(\mathrm{KX}+F\right)=\overset{•}{X}\;.$$

And $\overset{•}{X}$ can be expressed approximately as Equation (6):(6)$$\overset{•}{X}=\theta f\left(X_{n+1}\right)+\left(1-\theta \right)f\left(X_{n}\right),(0\leq \theta \leq 1).$$

In addition, using the first-order discretization of the first derivative:(7)$$\overset{•}{X}=\frac{X_{n+1}-X_{n}}{\Delta t_{n}}.$$

Results in
(8)$$X_{n+1}=X_{n}+{\Delta t}_{n}\overset{•}{X}=X_{n}+{\Delta t}_{n}\left(\theta f\left(X_{n+1}\right)+\left(1-\theta \right)f\left(X_{n}\right)\right).$$

Therefore, the rotation angle θ of the rotor at each time step will directly affect the accuracy and stability of the time-varying function solution. It is assumed that the angle of each rotation in the rotor is 7.2° in the simulation calculation process, and the total simulated rotation angle of the rotor is 360°, that is, the rotation process of the rotor is completed in 50 steps, to ensure the accuracy and stability of the calculation.

#### 2.3.2. Basic Equation

For polymer melt flow, the basic governing equation can be expressed as follows [20]:

The conservation of continuity equation:(9)$$\frac{\partial \rho }{\partial t}+\frac{\partial \left(\rho v_{x}\right)}{\partial x}+\frac{\partial \left(\rho v_{y}\right)}{\partial y}+\frac{\partial \left(\rho v_{z}\right)}{\partial z}=0.\;$$

The specific meaning of each unknown in Equation (9) is as follows:*ρ*—the density of the melt;*t*—time;*v_x_, v_y_, v_z_*—the component of the velocity vector in the x, y, z direction;

The conservation of momentum equation:

Momentum equation in the x direction
(10)$$\rho \left(\frac{\partial v_{x}}{\partial t}+v_{x}\frac{\partial v_{x}}{\partial x}+v_{y}\frac{\partial v_{x}}{\partial y}+v_{z}\frac{\partial v_{x}}{\partial z}\right)=\rho g_{x}-\frac{\partial p}{\partial x}+\left(\frac{\partial \tau_{xx}}{\partial x}+\partial \frac{\tau_{yx}}{\partial y}+\frac{\partial \tau_{zx}}{\partial z}\right);$$

Momentum equation in the y direction
(11)$$\rho \left(\frac{\partial v_{y}}{\partial t}+v_{x}\frac{\partial v_{y}}{\partial x}+v_{y}\frac{\partial v_{y}}{\partial y}+v_{z}\frac{\partial v_{y}}{\partial z}\right)=\rho g_{y}-\frac{\partial p}{\partial y}+\left(\frac{\partial \tau_{xy}}{\partial x}+\partial \frac{\tau_{yy}}{\partial y}+\frac{\partial \tau_{zy}}{\partial z}\right);$$

Momentum equation in the z direction
(12)$$\rho \left(\frac{\partial v_{z}}{\partial t}+v_{x}\frac{\partial v_{z}}{\partial x}+v_{y}\frac{\partial v_{z}}{\partial y}+v_{z}\frac{\partial v_{z}}{\partial z}\right)=\rho g_{z}-\frac{\partial p}{\partial z}+\left(\frac{\partial \tau_{xz}}{\partial x}+\partial \frac{\tau_{yz}}{\partial y}+\frac{\partial \tau_{zz}}{\partial z}\right).$$

The specific meaning of each unknown in Equations (10)–(12) is as follows:
P—the pressure on a fluid microelement;$\tau_{ij}$—the component of the adhesive stress τ acting on the surface of the microelement due to molecular viscosity;gx, gy, gz—the mass force of the microelement along the x, y, z axis.

The conservation of energy equation:(13)$$C_{v}\left(\frac{\partial T}{\partial t}+v_{x}\frac{\partial T}{\partial x}+v_{y}\frac{\partial T}{\partial x}+v_{z}\frac{\partial T}{\partial x}\right)$$

The specific meaning of each unknown in Equation (13) is as follows:T—temperature;C_v_—constant-volume specific heat;qx, qy, qz—the heat flux density of fluid flowing in unit time and unit area along the x, y, z axis direction.

### 2.4. Characteristics of Material Rheological Parameters

For the simulation, the BC (Bird-Carreau) model was used, which claims that the material exhibits the rheological properties of a pseudoplastic fluid at a high shear rate and a Newtonian fluid at a low shear rate [21]. The relationship between shear rate and viscosity can be expressed as follows:(14)$$\eta (\overset{•}{\gamma })=\eta_{\infty }+\left(\eta_{0}-\eta_{\infty }\right)\left(1+\left(\lambda \overset{•}{\gamma }\right){}^{2}\right){}^{\left(n-1\right)/2},$$
where η_0_ is the viscosity at zero shear rate (Pa·s), η_∞_ is the infinite shear viscosity (Pa·s), λ is the rubber viscoelastic characteristic time (s), and n is the Power-law index.

At 100 °C, the rheological curve of the experimental material was measured using a capillary rheometer RH-2000 produced by Malvern Company, in the UK, and fitted with the BC model. The specific values of the above physical quantities were obtained as follows: η_0_ = 1,423,133; η_∞_ = 0.0138; λ = 14.6707; n = 0.2590189. The rheological curve of the BC model material is shown in Figure 2.

### 2.5. Finite Element Meshing

CREO (a CAD (Computer Aided Design) design software package launched by PTC in Massachusetts, USA) was used to establish the three-dimensional model of the flow field and the rotor, which was subsequently meshed by ANSYS ICEM. The configuration of the flow channel remained unchanged, and the mesh only needed to be divided once in the entire simulation process. ANSYS POLYFLOW incorporates a technique known as mesh superposition to simulate transient flows with internal moving parts. MST grid technology was used to superimpose the rotor grid model and the runner grid model [22]. The flow channel area was divided into a structured grid and encrypted at the boundary layer. The grid size was 2 mm, and a prismatic layer grid was used near the wall with a total thickness of 2 mm and 5 layers were divided. The rotor body was divided into equal-sized tetrahedral unstructured meshes. The grid size was 2 mm, and the number of grids was 164,168. To verify the grid independence, the pressure at the upper middle position of the rotor was selected as a reference value, and different fluid mesh numbers were used for calculation. As shown in Figure 3, the calculation result stabilized after the number of grids in the fluid area reached 90,000. So, 89,376 fixed-flow grid cells and 97,734 nodes were generated. Figure 4 is a schematic diagram of a flow channel, a pair of rotors, and the superimposed grid.

## 3. Rotor Optimization Process and Results

### 3.1. Standard PSO Algorithm

Particles were randomly generated in the solution interval according to the constraints, and the optimal value was acquired by continuously iterating the optimization process. During each iteration, the position and velocity of the particles were updated at any time by transmitting information to each other. The particles constantly approached the best position they had experienced and the best position in the group [23]. The calculation formula is as follows:(15)$$v_{\mathrm{id}}^{n+1}={w\;v}_{\mathrm{id}}^{n}+c_{1}{\;\mathrm{rand}}_{1}^{n}\;\left({\mathrm{pbest}}_{\mathrm{id}}^{n}-x_{\mathrm{id}}^{n}\right)+c_{2}{\;\mathrm{rand}}_{2}^{n}\;\left({\mathrm{gbest}}_{\mathrm{id}}^{n}-x_{\mathrm{id}}^{n}\right),$$
(16)$$x_{\mathrm{id}}^{n+1}=x_{\mathrm{id}}^{n}+v_{\mathrm{id}}^{n+1}.$$

Here, w is the weight of inertia; when w = 0, the particle loses the memory of its speed;

c_1_ is a self-learning factor. When c_1_ = 0, it is called a selfless particle swarm algorithm, which will cause particles to quickly lose group diversity and fall into an endless loop;

c_2_ is a group learning factor. If c_2_ = 0, it is called a self-cognitive particle swarm algorithm. Because there is no information transferred and shared between groups, it will directly lead to a slower convergence of the algorithm. Therefore, selecting an appropriate learning factor can avoid falling into an infinite loop while ensuring the convergence speed. Generally, c_1_ = c_2_ = 2 is selected;

${\mathrm{rand}}_{1}^{n}$ and ${\mathrm{rand}}_{2}^{n}$ are random numbers of the interval [0,1], which increases the randomness of particles during flight;

and $x_{\mathrm{id}}^{n}$ represents the position of particle i in the nth iteration.

The basic steps of the PSO algorithm can be described as follows [24]:Determine the optimization interval, generate a certain number of particles, and initialize the speed and position of each particle in the particle swarm;Calculate the target vector of each particle and add the non-dominated solution to the external memory;Determine the initial best global position of the particles and the best position of each particle;Update the speed and position of the particles, and take certain measures to ensure the movement of particles in the optimization interval;Calculate the target vector value of each particle and adjust their individual best position;Update the external memory and select the global best position for each particle at the same time;Determine whether the target meets convergence conditions. If it does, the algorithm stops and enters the optimal value. If it does not, it returns to step 3 to recalculate.

The basic process is shown in Figure 5.

MATLAB software was used to write the corresponding optimization program. The specific program consisted of individual and group optimization, speed and position update, and out-of-bounds judgment programs.

### 3.2. Optimization Process and Results

The optimization objects were α of the short-wing helix, β of the long-wing helix, and ratio λ of long to short wing. The corresponding optimization intervals were: 32°≤α≤55°, 32°≤β≤46°, 1.8≤λ≤3.5, α≥β and the optimization objective was the global distribution index.

The objective function: $f\left(x\right)=\min(\int_{0}^{50}g\left(x\right)\mathrm{dt})$ was used to ensure the best global distribution and mixing effect of the mixer, that is, to ensure that the value of the global distribution index was the lowest in the 50 s mixing cycle. The objective function value was the integral value of the global distribution exponential function curve over time.

The specific values of the convex wing parameters of each generation are shown in Table 2. Because the total length of the rotor was fixed, and the long and short wings did not overlap, the length wing value was displayed instead of the long and short wing ratio values in the table. The fitness function values of each generation are shown in Table 3. It can be observed that when the number of iterations of the PSO algorithm is equal to 4, that is, after cyclic simulation of 24 rotor configurations, the parameters of the fifth-generation rotor configuration generated were stable at a long-angle spiral angle equal to 34°. The helix angle of the short wing was equal to 47°, the length of the long wing was 64 mm, and the length of the short wing was 29 mm, making the ratio of the long to short wing 2.2. At this time, the fitness function value distribution was relatively stable, and it was judged as convergent.

After optimization, the long-angled helix angle of the rotor did not change compared to the value before optimization, decreasing only by 1°. However, the short-angled helix angle increased by 12° compared to the one before optimization. In addition, the long wings were reduced by 8 mm, and the corresponding short wings were increased by 8 mm.

## 4. Analysis of Synchronous Rotor before and after Optimization

### 4.1. Comparison of Rotor Configuration

Table 4 shows a comparison between the convex wing parameters of the rotor before and after optimization.

According to the data in Table 4, the spiral angle of the long wing of the rotor after optimization did not change significantly compared with that before optimization, being reduced only by 1°. On the other hand, the spiral angle of the short wing changed greatly, being increased by 12° compared with that before optimization. In addition, the dimension of the long wings decreased by 8 mm, while the short wings increased by 8 mm.

Figure 6 shows the comparison of the expansion diagram of the two rotors, and Figure 7 shows the physical comparison of the rotors before and after optimization.

In Figure 6, the red line segment represents the unfurled wing of the rotor before optimization and the black line represents the unfurled wing of the rotor after optimization. It can be seen from the expansion figure that the flow channel between the long and the short wings of the rotor after optimization is significantly smaller than that before optimization, namely Q1 < Q2. Therefore, the flow rate of the adhesive when it passes through the intersection between the long and short wings is affected accordingly. Since the total mass of rubber and packing remained constant, the resistance of the rubber material when passing through the intersection between the long and short wings increased, the extrusion effect became more relevant, and the turbulence of the rubber material intensified, thus improving the mixing effect. When the colloidal volume at the intersection between the long and short wings is too large to pass through, most of the rubber will inevitably flow through the wing tip clearance, which belongs to the high shear zone. This process also contributes to the improvement of the mixing performance.

The helix angle of the long wing of the rotor after optimization is α=34°≤ϕ=34°~38°, φ is the friction angle between rubber and rotor, and the material cannot move in the axial direction on the long wing. The optimized short wing helix angle is β=47°>ϕ=34°~38°, and, under these conditions, the material can move in the axial direction on the short wing. The short wing continuously transports the material from the long wing to the wing and back to the center of the rotor for mixing, and the turbulence of the rubber material is strengthened in the reciprocating process, so it is more conducive to the mixing of the material.

### 4.2. Comparative Study of Contours

The pressure, shear rate, and mixed exponential fields in the flow field were comparatively studied to better understand the characteristics of the flow field of the synchronous rotor mixer before and after optimization. The axial center area of the rotor, that is, the cross-section with an axial distance of 46.5 mm, was selected as the reference to facilitate the comparison and description. The selected comparison times were 0.1, 0.5, and 0.9 s, with a rotation of 36°, 180°, and 324°, respectively.

#### 4.2.1. Pressure Cloud Analysis

Figure 8 compares the pressure cloud diagrams of the synchronous rotor mixer before and after optimization. A positive pressure area at the front peak surface of the rotor is observed in the entire mixer flow field, and a negative pressure area is formed at the rear peak surface. The maximum pressure of the rubber melt appears at the gap in the top of the rotor rib. Das et al. [25] simulated the rotor pressure field, and their research also reached similar conclusions. This also proves that the results of this model are valid. During the mixing process, the rubber will flow in the channel between the outer surface of the rotor and the inner wall of the mixing chamber. The rubber enters the high-shear area of the apex clearance from the front peak surface of the rotor, and then enters the rear peak surface area. During rotation, the volume space of the front peak surface gradually decreases, the rubber material is squeezed, the pressure increases, and the dispersed phase undergoes sorting and stretching along the flow direction. As the rotor rotates, the flow interface of the rubber is constantly shrinking, pressure increases, and the flow interface is attenuated to the minimum at the crest gap, where the pressure peak appears. In the drag region of the rotor, that is, at the rear peak surface, pressure decreases rapidly due to the sudden increase in the area of the flow channel.

By comparing the pressure cloud diagrams, it was found that the area of positive high pressure formed by the front peak surface of the rotor after optimization is larger than that before optimization, so a greater pressure gradient appears. This is because the radius of curvature of the rear peak surface in the optimized rotor is large, generating a large flow area at the rear peak surface, where the rubber polymer can quickly release pressure. Therefore, this condition is more effective for promoting the agglomeration of aggregates during the mixing process. Moreover, this hierarchical pressure gradient can realize the progressive change of rubber state, ensuring the quality of the rubber compound.

#### 4.2.2. Shear Rate Cloud Analysis

Figure 9 shows the comparison of the shear rate cloud diagrams of the synchronous rotor mixer before and after optimization. It can be observed that the maximum shear rate appears at the maximum diameter of the circle of rotation in both cases, and the minimum shear rate is distributed at the circumference of the base circle. This is because, during rotation, the gap between the turning circle and the inner wall of the mixing chamber is the smallest, and the rubber will be subjected to a stronger shearing effect when flowing through this area. At the circumference of the base circle of the rotor, the gap between the rotor and the rotor, and between the rotor and the inner wall of the mixing chamber, is relatively large. The polymer is in a relatively loose state in these regions, and the compression and shear effects are weak, so the shear rate is low. The maximum shear rate of the rotor after optimization is 336.3 s^−1^, and the maximum shear rate of the rotor before optimization is 291.5 s^−1^. This is because the pressure difference between the front and rear peak surfaces in the optimized rotor is greater during rotation, which increases the volume flow rate and linear flow velocity of the rubber at the crest gap. The increase of the shear rate has an important effect on the improvement of the dispersion and mixing ability of the internal mixer.

#### 4.2.3. Mixing Index Cloud Analysis

Figure 10 shows no significant differences between the rotor mixing index clouds before and after optimization. The mixing index near the circumference of the base circle of the rotor is between 0.2–0.4, which indicates that this area is dominated by rotational flow. Because the boundary condition is set to a non-slip boundary, the flow speed of the rubber material on the rotor surface is consistent with the linear velocity, so the rubber material will move close to the rotor surface, decreasing the mixing index in this region and appearing as pure rotational motion. The maximum mixing index is found in the middle of the two rotors near the upper and lower pins at a value of 0.8–1. This indicates that tensile flow occurs mainly in this area. Because the initial phase angles of the rotors are all arranged at 90°, the volume of the left and right half-cavity rotors near the middle position are not equal. During rotation, the rate of volume change at the upper and lower pins is high. Under the stirring action of the two rotors and the shunting effect of the upper and lower pins, the rubber will be strongly stretched and folded, exhibiting a higher mixing index.

### 4.3. Comparative Numerical Analysis

#### 4.3.1. Analysis of the Stretch Length

The logarithmic stretch length of the rotor before and after optimization is shown in Figure 11, and it can be observed that it increases linearly with time. In other words, the change in stretch length increases exponentially with time. Therefore, effective laminar mixing occurred in the flow field of the two rotors.

Moreover, the logarithmic tensile length curves corresponding to the optimized and unoptimized rotors are comparable in the first 15 s, after which the curve of the optimized rotor starts to deviate from the unoptimized rotor. As mixing time increases, the difference between the two curves becomes more pronounced. Therefore, the micro-facets in the optimized rotor mixer experience a higher stretch length and a better laminar mixing effect, resulting in better mixing performance.

#### 4.3.2. Analysis of Average Mixing Efficiency

As demonstrated in Figure 12, the mixing efficiency of both the optimized and unoptimized rotor is greater than 0, which means that the particles are always undergoing the stretching orientation process. When the particles start to move from standstill, the mixing efficiency instantly increases and then decreases rapidly, finally stabilizing after a certain mixing time. The time-averaged mixing efficiency of the optimized rotor has been improved, which means that the particles have a stronger tensile orientation effect in the flow field of the optimized mixer rotor.

#### 4.3.3. Analysis of the Global Distribution Index

When adding fillers, such as carbon black, to the mixer, since the position of the top plug is determined and the feeding process is fast, it can be assumed that the filler is more concentrated when entering the mixer, and the distance between particles is the smallest at this point. As mixing time increases, the particles migrate with the rotation of the rotor, and the distance between particles increases. When the mixing process reaches its end, the filler should be randomly and uniformly distributed in the mixing chamber in equipment with excellent mixing performance, and the global distribution index can be used to evaluate such characteristics.

The variation of the global distribution index with time for the rotor before and after optimization is shown in Figure 13. It can be observed that as mixing time increases, the global distribution indices of both rotors decreases and stabilizes after 25 s. This is because the rotor has completed an integer number of rotations within each statistical time, and the particles will complete the periodic cyclic migration movement with a high probability. However, this does not mean that the distribution mixing has been completed. When the initial number of particles is increased, the stabilization process becomes slower. In comparison, the global distribution index of the rotor after optimization is slightly smaller, which means that the distribution of particles in the entire mixing chamber is more uniform, and the distribution mixing performance of the mixer is better in this scenario.

#### 4.3.4. Separation Scale and Visual Analysis

The separation scale is a measure of the area where the concentration is homogenized, and its value decreases as the degree of mixing increases. When two particles reach a random uniform distribution, the separation scale is minimized. The change in the separation scale is not only affected by the size of the flow field but also depends on the initial distribution of the concentration and the number of tracking particles. Therefore, it is necessary to ensure that the number of particles and the initial distribution state is equal before doing a comparative analysis. If there is a dead mixing zone in the flow field, either the particles cannot reach the area or the particles originally distributed in the area cannot leave, and the corresponding separation scale cannot be minimized.

It can be seen from Figure 14 that the separation scale of the optimized rotor is greatly reduced and was stabilized in a short time. This suggests that the axial distribution mixing ability of the rotor after optimization is greatly improved compared with that before optimization. The distribution, mixing, and homogenization of materials generally rely on the axial reciprocating cutting action of the rotor. The continuous reciprocating movement of the rubber between the wings causes the material layer to be continuously updated and, finally, forms a uniform distribution of powder. According to the analysis, the optimized rotor greatly enhances the axial mixing ability of the mixer.

Ten thousand material particles without mass were released in the entire flow field at the initial moment to compare and analyze the mixing effect between the rotors before and after optimization, with no force between these particles. In addition, the concentration of each material particle needed to be defined. As shown in Figure 15, the red particles represent a particle concentration of 1, and the blue particles represent a particle concentration of 0. Figure 16 shows the particle distribution at 3, 10, and 40 s before and after optimization. It can be seen from the figure that the number of particles that migrated to the opposite half of the mixing chamber in the rotor before optimization is small at the different times, and both red and blue particles are mostly concentrated on the top pin of the mixer and the middle area of the two rotors. The optimized rotor exhibits both particles distributed more evenly with the particles migrating to the other half of the mixing chamber on a larger scale, while no phenomenon of concentrated distribution is observed. The visual analysis of the particles proves that the optimized rotor achieved the uniform mixing of particles more easily.

### 4.4. Comparative Experimental Study

#### 4.4.1. Experimental Raw Materials and Formulas

The specific materials and dosages used for the comparative experimental study are shown in Table 5. Since this experiment did not require a vulcanization process, the formulation did not contain the necessary components of the vulcanization system. Five groups of experiments were performed on the rotors before and after optimization.

#### 4.4.2. Experiment and Test Process

The experiment consisted of uniformly mixing the rubber with various fillers in the internal mixer and then adding the mixed rubber to the open mixer to form a sheet. The speed of the internal mixer was set to 60 r/min, the fill factor was set to 0.85, and the pressure of the top bolt was 0.60 Mpa. In addition, the cooling water temperature was set to 45 °C, and the mixing time was 200 s. Subsequently, 60 test samples from each of the five groups of rubber compounds were mixed to form 60 random samples, and then the Mooney viscosity, density, and carbon black dipersion of the test samples were analyzed. Among them, the density and carbon black dispersion of each sample were tested five times, and the median value was taken. The Mooney viscosity was measured by the Mooney viscometer model Premier MV produced by Alpha Company in Akron, Ohio, USA Density was measured with a hydrometer model Percisa XB 220A produced by the Precisa company in Zurich, Switzerland. The carbon black dispersion was tested with the carbon black dispersion meter model DisperGRADER produced by Alpha Company in Akron, OH, USA. Figure 17 shows the photos of the test samples. Figure 18 shows the photos of the testing equipment.

#### 4.4.3. Experimental Results and Analysis

The coefficient of variation was used to compare the degree of dispersion of the experimental data for the rotor before and after optimization to determine the stability of the data distribution. The coefficient of variation can be calculated as:(17)$$C.V=\frac{\sigma }{\mu },\;$$
where σ is the standard deviation, and μ represents the average.

Compared with the standard deviation, the coefficient of variation can eliminate the influence of the unit and the average value on the comparison of the degree of dispersion.

The data in Table 6. reveal that the coefficients of variation of the experimental samples of the optimized rotor are lower than those of the optimized front rotor in varying degrees. Therefore, according to the definition of the coefficient of variation, it can be determined that the test data distribution of the rotor after optimization is relatively stable, which proves that the material distribution is more uniform in the rotor after optimization. In addition, it can be observed that the rubber processed with the optimized rotor has a lower average Mooney viscosity and a higher carbon black dispersion. This demonstrates that the masterbatch compounded by the optimized rotor mixer has better processability. The dispersion effect on the filler is also improved, which is consistent with the results from simulation and optimization analysis.

## 5. Conclusions

POLYFLOW was used to carry out fluid simulation calculations for synchronous rotors with different configuration parameters. The rotor configuration was optimized using the PSO algorithm, taking the global distribution index of the mixer as the optimization parameter, and a comparative analysis of the rotor before and after the optimization was performed.

The parameterized design program of the rotor and the standard PSO optimization algorithm were written using MATLAB software. The results reveal that the length of the long wings was reduced by 8 mm, the helix angle was reduced by 1°, the length of the short wings was increased by 8 mm, and the helix angle increased by 12° compared with the rotor before optimization. The geometric model, cloud diagram, numerical analysis, and experimental studies of the rotors before and after optimization were compared. The results show that the optimized rotor is superior to the unoptimized rotor in the dispersion and distribution mixing effect. The numerical simulation optimization results are consistent with theoretical analysis and experimental results, which proves the feasibility of the numerical simulation optimization.

## Figures and Tables

**Figure 1 materials-13-05353-f001:**
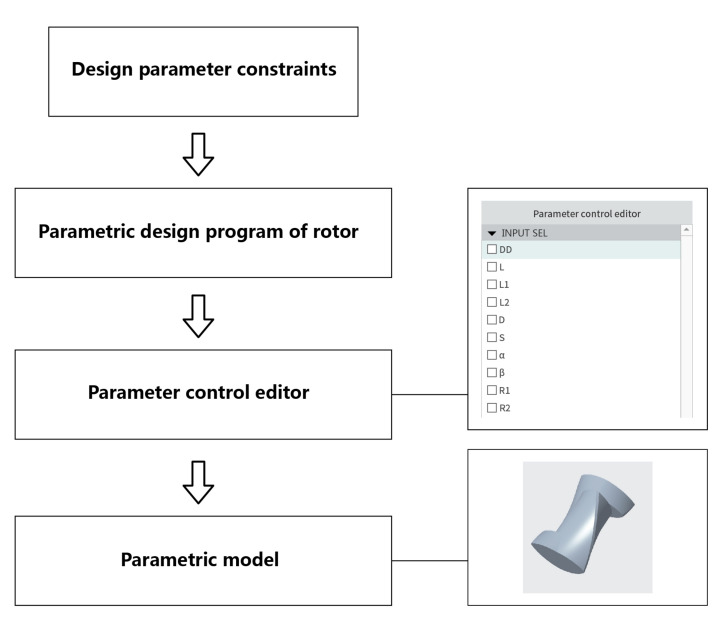
Design steps of the parametric model of the synchronous rotor.

**Figure 2 materials-13-05353-f002:**
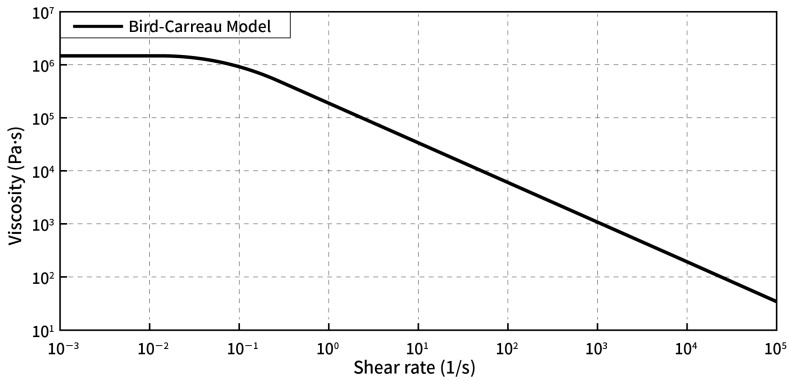
Rheological model of rubber.

**Figure 3 materials-13-05353-f003:**
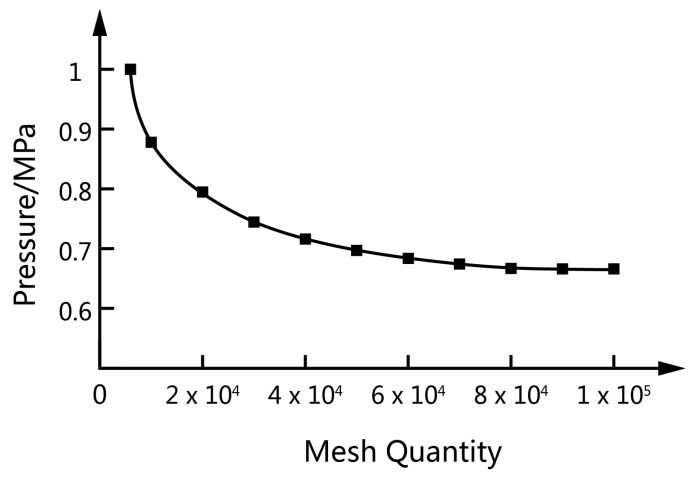
Mesh independence verification.

**Figure 4 materials-13-05353-f004:**
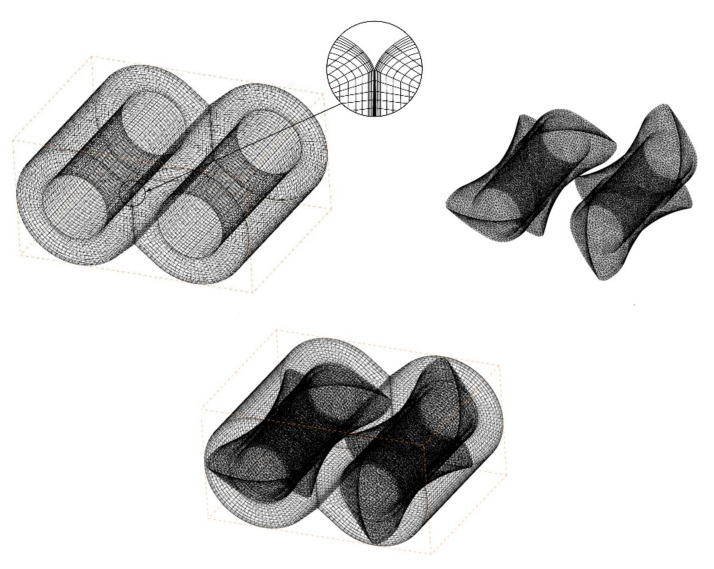
Finite element meshing with ICEM.

**Figure 5 materials-13-05353-f005:**
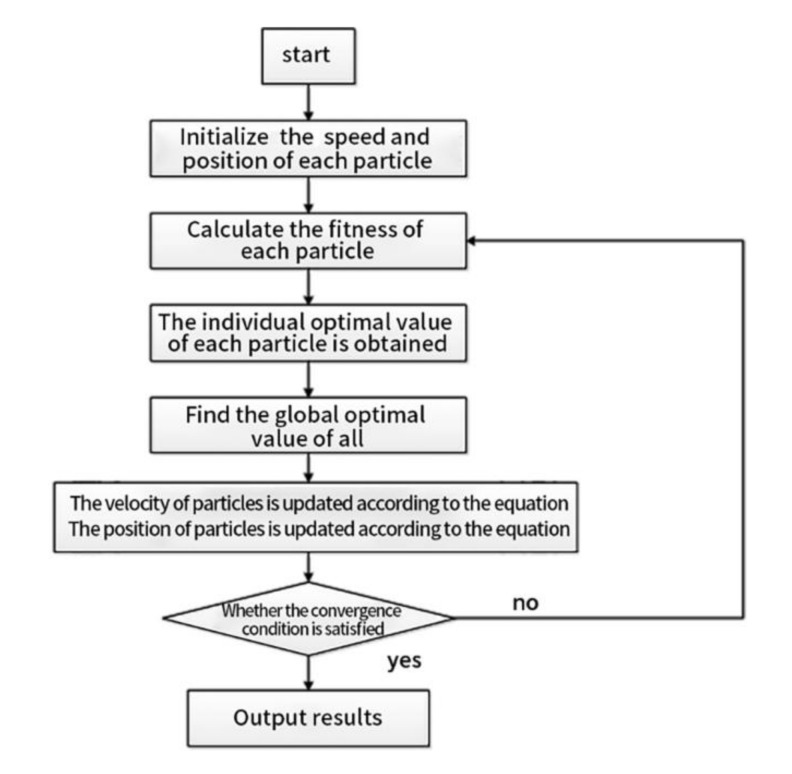
Flow chart of the standard particle swarm optimization (PSO) algorithm.

**Figure 6 materials-13-05353-f006:**
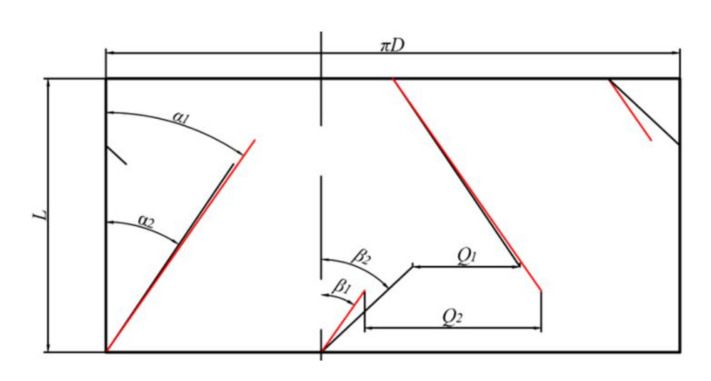
Comparison of internal mixer rotor expansion before and after optimization (diagram).

**Figure 7 materials-13-05353-f007:**
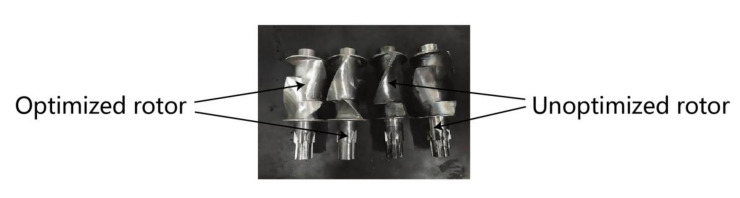
Comparison of internal mixer rotor expansion before and after optimization (photograph of specimens).

**Figure 8 materials-13-05353-f008:**
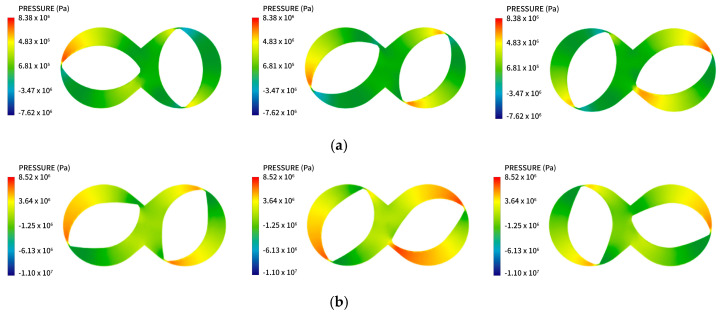
Pressure field of the rotor (**a**) before and (**b**) after optimization.

**Figure 9 materials-13-05353-f009:**
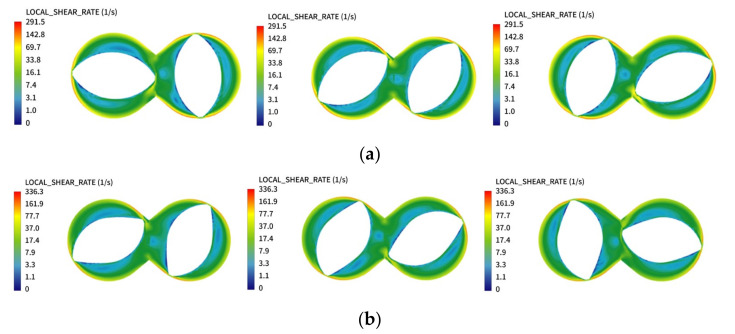
Shear rate field of the rotor (**a**) before and (**b**) after optimization.

**Figure 10 materials-13-05353-f010:**
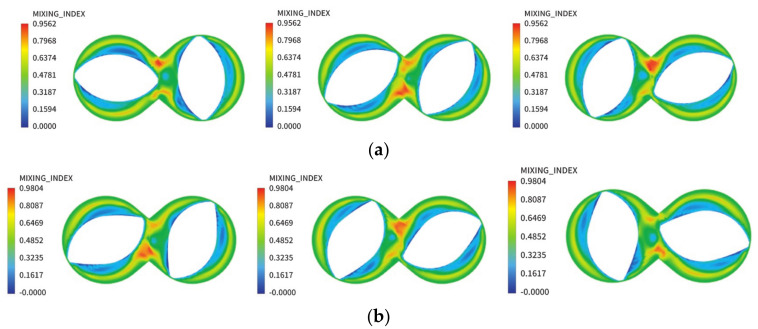
Mixed index field of the rotor (**a**) before and (**b**) after optimization.

**Figure 11 materials-13-05353-f011:**
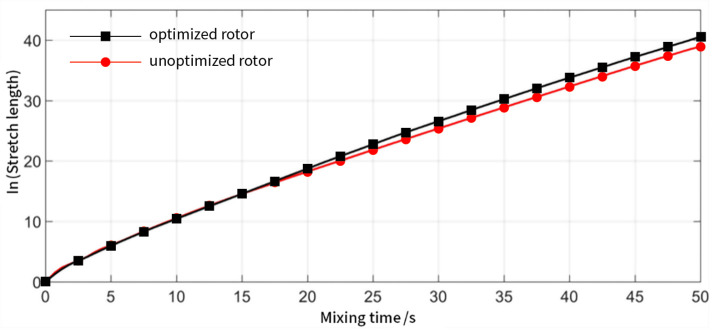
Mean logarithmic stretch length of the optimized rotor (black) and rotor before optimization (red) as a function of time.

**Figure 12 materials-13-05353-f012:**
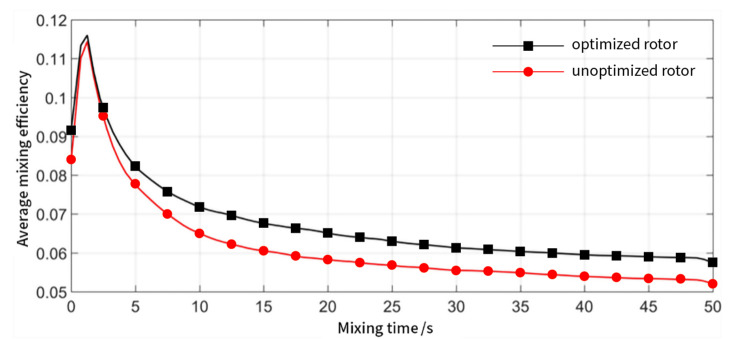
Average mixing efficiency of the optimized rotor (black) and rotor before optimization (red) as a function of time.

**Figure 13 materials-13-05353-f013:**
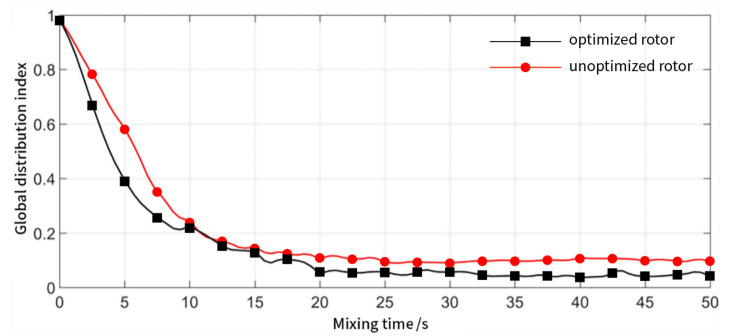
Global distribution index for the optimized rotor (black) and rotor before optimization (red) as a function of time.

**Figure 14 materials-13-05353-f014:**
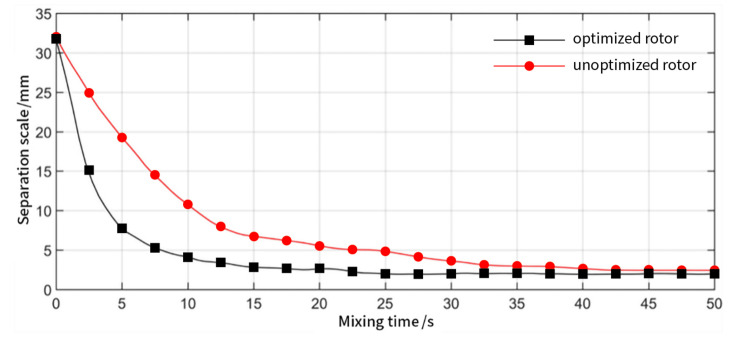
Separation scale as a function of time curve of distributed mixing between two internal mixer chambers for the optimized rotor (black) and rotor before optimization (red).

**Figure 15 materials-13-05353-f015:**
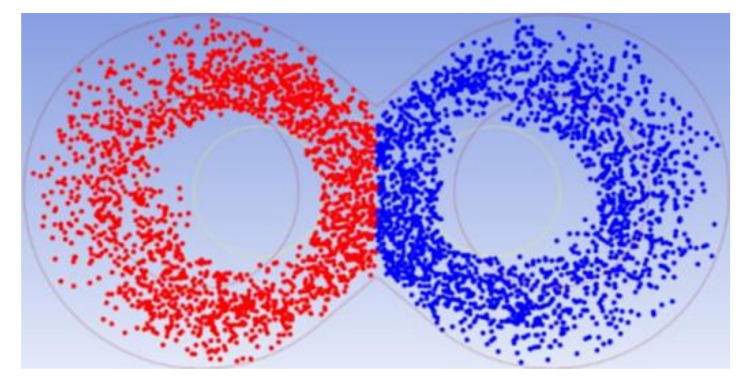
Initial particle distribution.

**Figure 16 materials-13-05353-f016:**
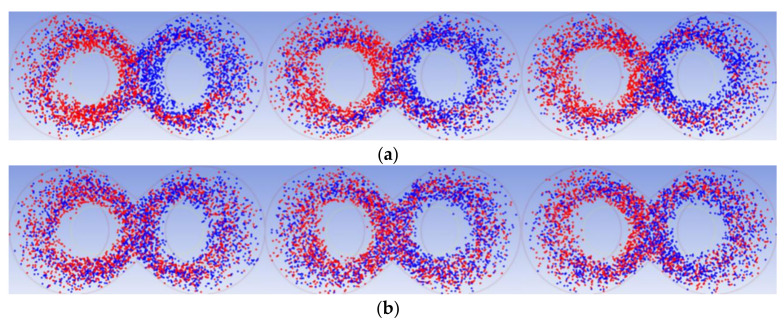
Particle distribution at 3, 10, and 40 s between the two chambers for the rotor (**a**) before and (**b**) after optimization.

**Figure 17 materials-13-05353-f017:**
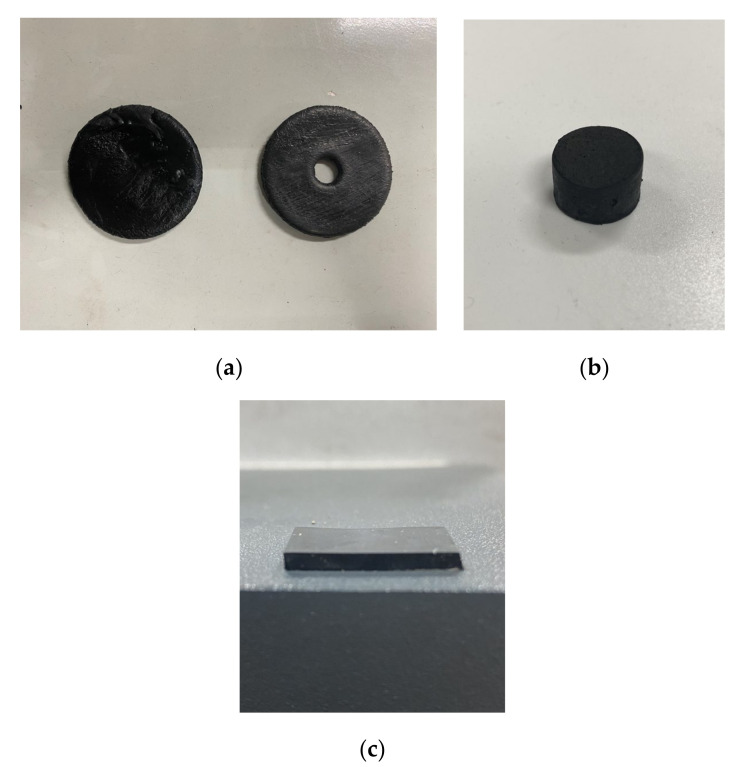
Mooney (**a**), density (**b**), and carbon black dispersion (**c**) test sample photos.

**Figure 18 materials-13-05353-f018:**
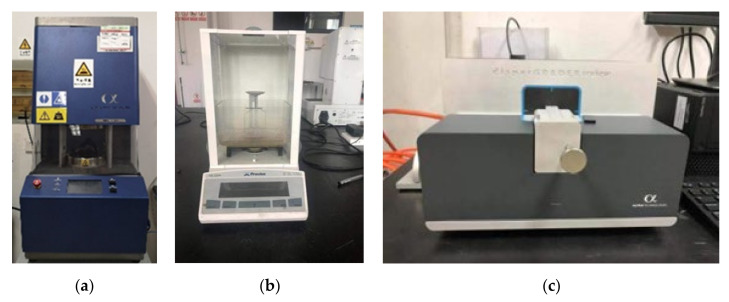
Photos of (**a**) Mooney viscosity, (**b**) hydrometer, and (**c**) carbon black dispersion meter.

**Table 1 materials-13-05353-t001:** Rotor design parameters.

Geometric Parameters	Parameter Values	Geometric Parameters	Parameter Values
Rotor wing width (mm)	3	Helix angle of short-wing (°)	Variable (Parametric control)
Gap between rotors (mm)	1	Maximum swivel circle diameter (mm)	62
Front surface radius (mm)	31	Ratio of long and short wings	Variable (Parametric control)
Back surface radius (mm)	100	Center distance of rotors(mm)	63
Base circle diameter (mm)	35	Axial length of rotor(mm)	93
Helix angle of long-wing (°)	Variable (Parametric control)		

**Table 2 materials-13-05353-t002:** Convex edge parameters for each generation of particles.

Group	Serial Number	Helix Angle of Long Wing (°)	Helix Angle of Short Wing (°)	Length of Long Wing (mm)	Length of Short Wing (mm)
1	1	41	46	68	25
	2	40	53	72	21
	3	34	47	64	29
	4	41	46	72	21
	5	35	39	64	29
	6	37	46	68	25
2	1	36	46	66	27
	2	36	52	71	22
	3	34	47	64	29
	4	35	47	68	25
	5	34	44	64	29
	6	37	47	67	26
3	1	37	47	66	27
	2	33	47	70	23
	3	34	47	64	29
	4	33	47	66	27
	5	34	46	64	29
	6	35	48	67	26
4	1	36	47	66	27
	2	33	45	67	26
	3	34	47	64	29
	4	34	47	66	27
	5	34	47	64	29
	6	34	48	66	27
5	1	34	46	65	28
	2	35	46	65	28
	3	34	47	64	29
	4	34	47	65	28
	5	34	47	64	29
	6	34	47	64	29

**Table 3 materials-13-05353-t003:** Fitness function values for each generation of particles.

	Group	1	2	3	4
Serial Number					
1		12.8196	10.7240	11.2609	10.8838
2		13.4446	12.5034	12.8554	10.9776
3		10.5061	10.5061	10.5061	10.5061
4		12.7168	12.3610	10.5796	10.7224
5		11.5913	11.0641	10.7576	10.5061
6		12.8548	12.3343	12.7344	10.7512

**Table 4 materials-13-05353-t004:** Rotor geometric parameters before and after optimizing.

	Rotor Type	Synchronous Rotor before Optimization	Synchronous Rotor after Optimization
Ridge Parameters			
Helix angle of long wing (°)		35	34
Helix angle of short wing (°)		35	47
Length of long wing (mm)		72	64
Length of short wing (mm)		21	29

**Table 5 materials-13-05353-t005:** Raw materials and dosage for experimental study.

Material Name	Phr	Material Name	Phr
RC2557S	110	Silical115MP	65
SBR1723	30	Si69 mix	8.4
ZnO	2	N234	25
SAD	2	Antilux111	1.8
4020	2	V700	4

**Table 6 materials-13-05353-t006:** Experimental data from the synchronous rotor before and after optimization.

Serial Number	Mooney of Optimized Rotor (ML_1+4_)	Mooney of the Rotor Before Optimization (ML_1+4_)	Density of the Optimized Rotor (g/cm^3^)	The Density of the Rotor Before Optimization (g/cm^3^)	Carbon Black Dispersion of the Optimized Rotor	Carbon Black Dispersion of the Rotor Before Optimization
1	87.16	92.16	1.109	1.134	8.2	7.4
2	81.46	88.12	1.135	1.185	8.4	7.2
3	84.15	91.67	1.104	1.125	7.9	7.9
4	84.29	89.64	1.149	1.158	8.3	7.1
5	81.49	87.4	1.143	1.194	8.4	8.1
6	86.78	88.87	1.156	1.105	7.9	7.2
7	86.65	91.23	1.130	1.105	8	7.3
8	83.75	84.69	1.139	1.102	8.1	7.8
9	85.98	86.56	1.153	1.168	8.4	7.2
10	81.87	92.23	1.132	1.160	8.2	7.6
11	84.93	86.62	1.117	1.111	8	7.9
12	82.59	89.06	1.159	1.180	8.2	8.1
13	83.8	91.36	1.102	1.162	8.2	7.9
14	86.39	92.26	1.120	1.107	8.1	7.9
15	82.18	89.01	1.158	1.107	7.8	7.5
16	82.88	83.83	1.122	1.114	8.3	7.3
17	84.04	83.14	1.119	1.179	8.2	7.6
18	83.1	88.67	1.107	1.109	8.1	7.3
19	86.09	88.19	1.155	1.124	7.8	8
20	87.32	91.94	1.108	1.124	8.1	7.8
21	81.93	89.89	1.120	1.111	8	8
22	82.44	89.09	1.154	1.186	7.9	8
23	85.46	87.77	1.130	1.170	8.4	7.6
24	83.34	85.27	1.137	1.173	8.1	7.4
25	87.23	91.94	1.135	1.165	7.8	7.8
26	87.22	87.93	1.142	1.152	8.1	8.2
27	85.08	83.09	1.102	1.133	8.3	7.9
28	86.49	89.41	1.132	1.166	7.9	8
29	83.51	87.74	1.102	1.112	8.4	7.8
30	85.01	83.36	1.150	1.115	8	7.7
*μ*	84.49	88.4	1.131	1.141	8.1	7.7
*σ*	1.92	2.86	0.0189	0.0306	0.19	0.33
*C.V*	2.28%	3.23%	1.67%	2.68%	2.31%	4.24%

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
