# Peer review of "Numerical Optimization Simulation of Synchronous Four-Wing Rotor"

_materials, 2020, doi:10.3390/ma13235353_

Round 1
Reviewer 1 Report
This paper investigated on a method for optimizing the rotor structure by using optimization algorithms combined with numerical simulations. However, there are many drawbacks in the present manuscript. In the following, the revisions are mentioned:
- Punctuation should be applied correctly in the text.
- In the number reference system, the references should be listed in numerical order as they appear in the text. For example, in the text of the article, reference number [5] is used earlier than reference number [3].
- In the introduction section, more recent research should be used. Here are some suggestions regarding the multi-objective optimizations and optimization of rotors:
- Numerical investigation of the savonius vertical axis wind turbine and evaluation of the effect of the overlap parameter in both horizontal and vertical directions on its performance. Symmetry, 2019, 11(6), p.821.
- Multi-objective optimization of a pitch point absorber wave energy converter. Water, 2019, 11(5), p.969.
- All the references should be cited in the text. Reference [7] is not cited anywhere in the text.
- Please clearly explain in the “introduction” section that what the novelties of this research are.
- The quality of figures are very low. The texts in the figures are unreadable. Replace the figures with high resolution ones.
- Equation (1) is wrong. “” should be replaced with .
- Please clearly explain that how Equation (5) is obtained.
- The section number of “Basic equation” is wrong. Please modify it.
- All parameters in the equations should be defined after the presented equations. For example, parameters used in Equation (7-9) are not defined. Please check all the equations carefully and define all parameters.
- In line numbers 150 and 162, what “BC” and “CREO” stand for? All abbreviations should be defined in the text when they used for the first time.
- In line numbers 154 and 156, for “η0” and “η∞”, “0” and “∞” should be written as subscript.
- In figure 2 is provided from another reference, it should be cited properly.
- In Fig. 2, the unit of the vertical axis (viscosity) should be “Pa.s” instead of “Pa/s”.
- In section 2.5, please explain that what is “MST grid” technology?
- The quality of Fig. 3 is very low. I cannot recognize that how you meshed your model. You can zoom different parts to enlarge them in order to see the meshes vividly. Also, please explain that how did you mesh your model? What are the sizes of elements you used for the meshing? How did you meshed the model near the walls?
- The mesh independence study in this paper in very incomplete. For at least one parameter, draw a graph with the results of different amounts of meshes to assure us of selecting the best mesh size with acceptable accuracy.
- A validation study should be implement. You should compare the results of your model with a well-known study or experimental results in order to assure the researchers of the correctness and validity of your results.
- In line numbers 214, “46” should be replaced with “”.
- The title used for “Table 2” is not suitable. Please modify it.
- In section 3 and 3.2, although “analysis of results” is used in the titles, no analysis is used in these sections. Therefore, the titles should be modified.
- Use line legends in Figs. 10-13 to define the graphs.
- Section “4.4.2 Experiment and test process” is very brief. The experimental tests are not explained properly. Please show the real picture of test setup with the sample. Also, you should mention the model of sensors used for the measurements with their precisions.
- There are many vague points about your experimental procedure. Please clarify that: How many times did you carry out each test in order to assure the repeatability of the tests? 2. How did you validate your test results? At least, a sample of uncertainty analysis for a series of data should be presented.
Reviewer 2 Report
The publication is a comprehensively written report. It discusses the optimisation of geometrical parameters of a four-wing rotor on the basis of numerical calculations using the finite element method. In itself this aspect is not completely new. However, the publication correlates the numerical results with experimental results on modified rotors. This aspect seems quite original and interesting. All in all, the paper provides an interesting application of numerical simulations for virtual optimization processes.
For a simulation, simplifications must certainly always be made. However, it is suggested to critically discuss the assumptions listed in lines 106 - 111. Some of the simplifications are common and almost uncritical, but others can have an disadvantageous influence on the results and do not completely correspond to the real conditions. It would also be interesting to learn more about the selected finite element type and the numerical CFD solution procedure.
corrections:
Formula (1) : K(X)X
Formula (8): vj , j is an index!
Author Response
Thank you for your review. I have revised the errors in formula (1) and formula (8). Regarding the assumptions listed in lines 106-111, I completely agree with you. The first and second assumptions are not completely correspond to the real conditions. At that time, considering the limitation of calculation speed, the assumptions were made. In the next research, I will consider adopting a more realistic assumption. And learn and choose more finite element types and numerical CFD solutions.
Round 2
Reviewer 1 Report
The paper is modified and improved considerably compared to its previous version. However, still some of modifications are not implemented in the revised paper which I mentioned them in the following:
- Please clearly explain in the “introduction” section that what the novelties of this research are.
- The quality of figures are very low. The texts in the figures are unreadable. Replace the figures with high resolution ones. Fig. 1 is not modified.
- If figure 2 is provided from another reference, it should be cited properly.
- A validation study should be implemented. You should compare the results of your model with a well-known study or experimental results in order to assure the researchers of the correctness and validity of your results.
Author Response
Thanks for your revise.
Please see the attachment.
